# MaskInversion: Localized Embeddings via Optimization of Explainability Maps

**Walid Bousselham**
Tuebingen AI Center
University of Tuebingen

**Sofian Chaybouti**
Tuebingen AI Center
University of Tuebingen

**Christian Rupprecht**
University of Oxford

**Vittorio Ferrari**
Meta

**Hilde Kuehne**
Tuebingen AI Center
University of Tuebingen
MIT-IBM Watson AI Lab

## Abstract

Contrastive vision-language foundation models have achieved tremendous results in global vision-language alignment, but still show some limitations in creating representations for specific image regions. To address this problem, we propose MaskInversion, a method that leverages the feature representations of pre-trained foundation models such as CLIP to generate a context-aware embedding for a query image region specified by a mask at test time. MaskInversion starts with initializing an embedding token and compares its explainability map, derived from the pretrained model, to the query mask. The embedding token is then subsequently refined to approximate the query region by minimizing the discrepancy between its explainability map and the query mask. During this process, only the embedding vector is updated, while the underlying foundation model is kept frozen allowing to use MaskInversion with any pre-trained model. As deriving the explainability map involves computing its gradient, which can be expensive, we propose a gradient decomposition strategy that simplifies this computation. The learned region representation can be used for a broad range of tasks, including open-vocabulary class retrieval, referring expression comprehension, as well as for localized captioning and image generation. We evaluate the proposed method on all those tasks on several datasets such as PascalVOC, MSCOCO, RefCOCO, and OpenImagesV7 and show its capabilities compared to other SOTA approaches[1].

## 1 Introduction

Foundation models such as CLIP (Radford et al., 2021), pre-trained with a contrastive loss on large-scale image-text datasets, have significantly advanced vision-language understanding. However, those models focus on a global vision-language alignment in training, matching the respective text and image class ([CLS]) tokens, thus only the globally pooled information. As a result, such models often struggle with tasks requiring precise localization or the recognition of specific image regions, necessitating novel approaches to harness their full potential. In the following, we tackle the problem of generating embeddings localized to specific image regions from pretrained vision-language models. While it is possible to obtain such embeddings via naïve solutions, e.g. by processing only the cropped region, or aggregating the local token embeddings over a mask, such simple approaches often do not yield optimal results: cropping can remove important context, while token aggregation over region features might not result in a good, aligned representation as local tokens do not always correspond to the correct representation (Zhou et al., 2022). Different approaches have been proposed to address the problem of localized vision-language tasks: ReCLIP (Subramanian et al., 2022) uses colored boxes during training to localize the alignment between vision and language. FGVP (Yang et al., 2023) employs different masking strategies to force the model to focus on the relevant object region. AlphaCLIP (Sun et al., 2024) finetunes CLIP together with an alpha channel to highlight

---

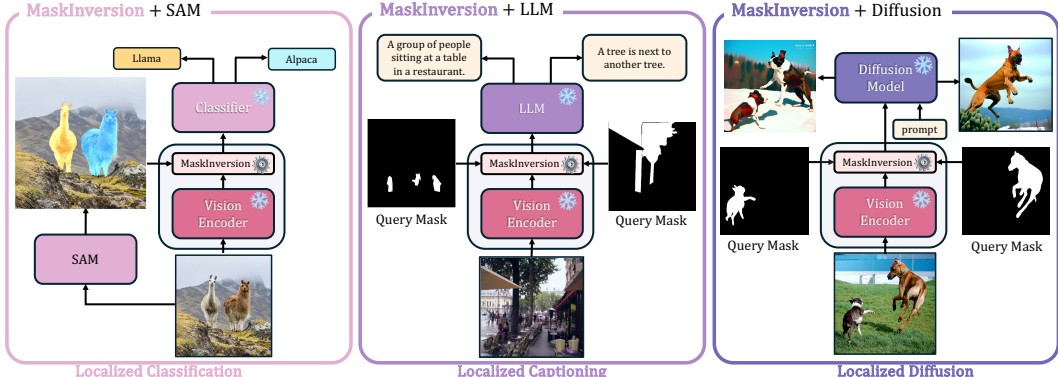

Figure 1: **MaskInversion Applications:** The proposed MaskInversion method generates a localized embedding without modifying the vision encoder, thereby enabling seamless integration as a drop-in replacement for the vision encoder output across various scenarios, such as *Localized Classification* to classify a specific region of an image, *Localized Captioning* to direct the attention of an LLM to specific parts of an image, or *Localized Diffusion* where the embedding is used in conjunction with a diffusion model to generate variations of specific regions of images.

the region of interest. Finally, RIS (Yu et al., 2023) proposes a token masking pipeline to achieve zero-shot referring image segmentation.

Following this line of works, we propose MaskInversion as a method to learn a localized embedding for a query image region specified by a mask at test time, see Figure 2. MaskInversion differs from previous methods as it does not adapt the vision-language backbone, but instead leverages the explainabilty map of a frozen backbone at test-time to optimize a representation, namely a token that captures the localized embedding for a given region mask. We start by initializing this localized embedding token from the global [CLS] token produced by CLIP. This token representation is then used to compute the initial explainability map for its current representation. We then compute the difference between the explainability map and the query mask and subsequently update the token so that its representation generates an explainability map that matches the query mask. As a result, we learn a token representation specific to the image region covered by the query mask. Note that the token representation learning process is done for each mask separately. Thus, several different localized embedding tokens are created from the same image when multiple object masks are given. We can enhance the computational efficiency in this case by exploiting the fact that the derivation of the explainability map is fixed because of the frozen backbone, and is independent of a query mask. Namely, we propose a gradient decomposition strategy that simplifies the gradient computation associated with the explainability method. Finally, while the resulting localized embedding tokens are optimized for their specific mask, it can sometimes be desirable to also include global context. We therefore propose an add-on regularization loss that aligns the learned representation to the global image representation and allows to balance between global and local representations.

The localized embeddings can be used in various downstream tasks, including region-based localized classification, region-based localized captions, and localized image generation (Figure 1). In all cases, we assume a zero-shot setting and use our localized embedding tokens as a drop-in replacement, e.g. for the CLIP ViT [CLS] token. This means e.g. for region-based zero-shot classification that we compute the localized embedding token and match it with the respective class prompts, e.g. "A photo of a dog". We evaluate the proposed method in all those scenarios, showing improved performance compared to other methods in each domain.

We summarize the contributions of this work as follows: (1) Given an image and a query mask, we learn a localized embedding at test time that captures the region characteristics within the mask in a single token. The learned token can be used as a drop-in replacement for any application based on the same backbone. (2) We propose gradient decomposition to make the process computationally efficient for multiple query masks in the same image. (3) We evaluate the resulting representation on various region-based downstream tasks, showing improved results across a range of different applications ranging from referring expressions to class retrieval and localized captioning.

## 2 RELATED WORK

**Localized Representation Learning.** The task of enhancing the localized embedding of foundation models such as CLIP (Radford et al., 2021) has gained increased attention recently. Various strategies have been proposed to leverage and enhance those backbones for localized vision-language tasks. For instance, ReCLIP (Subramanian et al., 2022) uses a combination of clipping and blurring to receive a region-specific embedding and further tries to capture relations between those instances. Shtedritski *et al.* (Shtedritski et al., 2023) found that a red circle around an object can direct the model's attention to that region, thus producing a 'localized' [CLS] token while maintaining global information. As an extension to those works, Yang et al. (Yang et al., 2023) explore different techniques for Fine-Grained Visual Prompting (FGVP), including outlining the relevant object or blurring the rest of the image (Blur Reverse Mask) and using the resulting CLIP CLS token for various downstream tasks. We find that especially the masked blurring provides a strong baseline. Another line of work, CPT (Yao et al., 2024) fine-tunes an existing language model to allow for a prompting based on different color patches. AlphaCLIP (Sun et al., 2024) takes a similar approach by retraining CLIP to take an alpha mask alongside the original image as input, focusing the model's output feature representation on the area covered by the alpha mask. However, this method requires millions of mask annotations to generalize effectively. Note that MaskInversion differs from both streams of work. In contrast to current visual prompt tuning methods (Shtedritski et al., 2023; Yu et al., 2023; Yang et al., 2023) , it does not seek to change the input image directly to get a localized CLS token embedding, but instead learns a new representation for the given mask. In contrast to methods that rely on masked-based pretraining Sun et al. (2024); Yao et al. (2024), MaskInversion is applied at test time and does not assume any adaptation of weights of the frozen backbone. Finally, Gal et al. (Gal et al., 2023) proposed text inversion as an idea to capture embeddings in a token that represents a certain object to be injected into a text-to-image generator. While serving as inspiration for this work, MaskInversion differs from text inversion as it captures regional properties via binary masks and respective explanation maps, whereas text inversion focuses on learning general object properties from multiple images.

**Explainability Methods.** Gradient-based methods, which compute explanations based on the gradient of the model's prediction with respect to the model output, are computationally efficient since they are a direct function of the model's parameters and do not rely on additional models or image modifications. They have been used successfully to identify reasoning, spurious correlation, and trustworthiness in traditional computer vision models (Erhan et al., 2009; Simonyan et al., 2014; Springenberg et al., 2015; Sundararajan et al., 2017; Selvaraju et al., 2017; Smilkov et al., 2017; Kapishnikov et al., 2019). Furthermore, gradient-based methods are differentiable, making it possible to use them as an objective function. (Chefer et al., 2022) uses the explainability map to supervise the model training, enforcing the model to base its classification prediction on the part of the image that contains the object, thus enhancing the model's robustness. Similarly, (Paiss et al., 2022) leverages the explainability signal to force an image generation model to utilize the entirety of the text prompt given by the user. Early explainability methods were specifically developed for Convolutional Networks, *e.g.* GradCAM (Selvaraju et al., 2017), GradCAM++ (Chattopadhay et al., 2018) and Integrated Gradients (Sundararajan et al., 2017). However, the widespread use of Vision Transformers (ViT) has led researchers to adapt or develop methods specifically for transformers. Rollout (Abnar & Zuidema, 2020) combines all the attention maps via matrix multiplication to trace the flow of importance through the transformer's layers. Chefer et al. (Chefer et al., 2021) extended rollout by weighting the attention by their gradient, making the method class-specific. Recently, LeGrad (Bousselham et al., 2025) proposed a gradient-based feature-attribution method specifically designed for ViT architectures relying on the gradient of the attention maps, making it fast and easy to use. We choose LeGrad as the default explainability method used in the evaluation, but note that MaskInversion is a general method and can be used with any differentiable explainability method.

## 3 METHODOLOGY

The proposed method, coined as *MaskInversion*, aims to learn a localized embedding or feature vector that encapsulates an object's characteristics within an image specified by a query mask.

As shown in Figure 2, our method starts with the initialization of an embedding vector that serves as the localized embedding token for the mask. This vector is iteratively refined through an optimization process guided by an explainability map which highlights the image regions most influential on the

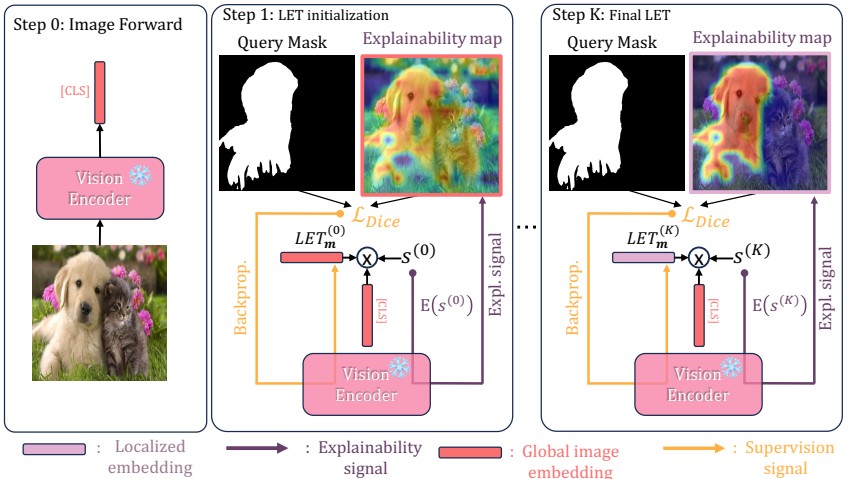

Figure 2: **Overview of the proposed method:** *Step 0*: the input image is forwarded only once during the whole MaskInversion process. *Step 1:* the localized embedding token $LET_\mathbf{m}$ is initialized by the vision encoder's [CLS] token. The $LET_\mathbf{m}$ is then trained such that its explainability map correlates to the query mask. *Step K:* after $K$ gradient descent steps, we obtain the final localized embedding $LET_\mathbf{m}$ that can be used for downstream tasks.

embedding, enabling targeted refinement. The optimization is supervised by enforcing similarity between the generated explainability map and the query mask. Optionally, a regularization loss can be applied to enforce the mask embedding to align with the model's learned manifold. Finally, we propose a gradient decomposition strategy to enhance computational efficiency, particularly when computing multiple embeddings for different masks on the same image.

## 3.1 PRELIMINARIES: EXPLAINABILITY METHODS

The proposed method relies on the use of explainability methods to guide the creation of the localized embedding token. Here, we give a brief introduction to explainability methods, focusing on "gradient-based" methods (e.g. GradCAM (Selvaraju et al., 2017)).

Let $\mathcal{F}$ denote a model that maps an input image $x \in \mathbb{R}^{3 \times W \times H}$ to an output activation $\mathcal{F}(x) = s \in \mathbb{R}$, where $s$ can be derived from a classifier's score or the cosine similarity between image and text embeddings in a vision-language model (e.g. CLIP). We denote $\mathbf{A}^l$ the intermediate representation of $\mathcal{F}$ at a given layer $l \in \{1, \dots, L\}$. $\mathbf{A}^l$ can be intermediate features maps in the case of CNNs (Selvaraju et al., 2017), intermediate tokens or attention maps in the case of ViTs (Dosovitskiy et al., 2021). We also denote the partial derivative of the activation $s$ w.r.t $A^l$ as $\nabla A^l = \frac{\partial s}{\partial A^l}$.

A gradient-based explainability method can be generally formulated as a combination of operations between the intermediate representation $\mathbf{A} = (A^1, \dots, A^L)$ and the gradients $\nabla \mathbf{A} = (\nabla A^1, \dots, \nabla A^L)$. It produces a 2D heatmap, denoted $E = g(\mathbf{A}, \nabla \mathbf{A}) \in \mathbb{R}^{W \times H}$. For instance, in GradCAM (Selvaraju et al., 2017), $E$ is defined as $E(\mathbf{A}, \nabla \mathbf{A}) = \text{ReLU}\left(\sum_k \alpha_k \cdot \mathbf{A}_k^L\right)$, where $\alpha_k = \sum_{ij} \nabla A_{k,i,j}^L$ are the weights for the feature maps $\mathbf{A}^L$. In the context of ViTs, we employ LeGrad (Bousselham et al., 2025), which considers the intermediate representations $\mathbf{A}^l$ to be the attention maps of the self-attention layers. For a given activation score $s$, the gradient $\nabla \mathbf{A}^l$ of $s$ with respect to the attention map $\mathbf{A}^l$ is computed, and a ReLU function is applied to discard negative contributions. While LeGrad averages the explainability maps of several layers, we here utilize only the attention map of the last layer to reduce computational cost.

## 3.2 LOCALIZED EMBEDDING LEARNING VIA EXPLAINABILITY MAP OPTIMIZATION

We denote the input for the proposed method as an image $x \in \mathbb{R}^{3 \times W \times H}$ and a binary query mask $\mathbf{m} = (m_{i,j}) \in \mathbb{R}^{W \times H}$, $m_{i,j} \in \{0, 1\}$, specifying a region of interest. Our objective is to derive a localized embedding token $LET_\mathbf{m} \in \mathbb{R}^d$ that generates an explainability map that corresponds to the masked region.

**Embedding Token Initialization.** We initialize the localized embedding token $LET_{\mathbf{m}}^{(0)}$ by copying the global [CLS] token produced by the foundation model, $LET_{\mathbf{m}}^{(0)} = z^0 \in \mathbb{R}^d$ (Step 0 in Fig.2). We then compute the cosine similarity between the embedding token and the average of the [CLS] and all patch tokens as the activation score for the explainability map (Step 1 in Fig.2)

$$s^{(0)} = \cos\left(LET_{\mathbf{m}}^{(0)}, \bar{\mathbf{z}}\right) \in \mathbb{R}, \tag{1}$$

where $\bar{\mathbf{z}} = \frac{1}{n}\sum_p z_p$ represents the combined patch and [CLS] token representation averaged across the spatial dimensions, and **cos** denotes the cosine similarity. The resulting similarity score is used to compute the explainability map denoted as $\mathbf{E}^{(0)} = E(s^{(0)}) \in \mathbb{R}^{W \times H}$, with each element $\mathbf{E}_{i,j}^{(0)} \in [0,1]$. This map $\mathbf{E}^{(0)}$ indicates the regions within the image that the initial embedding $LET_{\mathbf{m}}^{(0)}$ predominantly focuses on. Since the localized embedding is initialized with the [CLS] token our initial explainability map corresponds to the explainability map of the [CLS] token.

**Embedding Token Optimization.** To refine the initial estimate and guide the embedding token representation towards the query mask, we treat the mask localized embedding $LET_{\mathbf{m}} \in \mathbb{R}^d$, corresponding to the query mask $\mathbf{m}$, as a *learnable vector* with $d$ parameters. We supervise the learning of this vector by optimizing its parameters so that the resulting explainability map $\mathbf{E}^{(k)}$ for this token resembles the query mask $\mathbf{m}$.

We achieve this goal via iterative gradient descent. Specifically, we quantify the discrepancy between the explainability map and the query mask using a soft Dice loss, as commonly employed in segmentation tasks (Milletari et al., 2016; Cheng et al., 2021) for measuring region similarity:

$$\mathcal{L}_{\text{Dice}} = 1 - \frac{2 \times \text{intersection}(\mathbf{E}^{(k)}, \mathbf{m})}{\text{union}(\mathbf{E}^{(k)}, \mathbf{m}) + \epsilon}, \tag{2}$$

where $\text{intersection}(\mathbf{E}^{(k)}, \mathbf{m})$ and $\text{union}(\mathbf{E}^{(k)}, \mathbf{m})$ are the intersection, realized by elementwise multiplications, and union, by elementwise addition, of the explainability map and the binary mask; $\epsilon$ is a small constant to avoid division by zero. We minimize the Dice loss by optimizing the localized embedding $LET_{\mathbf{m}}$ parameters over $K$ iterations of gradient descent to achive the final embedding $LET_{\mathbf{m}} = LET_{\mathbf{m}}^{(K)}$ (see Step $K$ in Fig.2).

**Handling Multiple and Overlapping Masks.** It is worth noting that the optimization process described above is instantiated independently for each query mask. Consequently, *MaskInversion* naturally handles images containing dense object clusters or overlapping masks without cross-mask interference. Since the Dice loss (Eq. 2) is computed solely between the generated explainability map and the specific binary query mask, the presence of other objects or overlapping regions does not affect the convergence or quality of the target embedding.

**Regularization Loss.** The method as described so far will capture the representation of the region indicated by the query mask. This can cause the final representation $LET_{\mathbf{m}}$ to be less aligned with the global context of the overall image. However, it can be helpful to have both a good region representation together with global image context. We therefore introduce an add-on auxiliary regularization loss that forces the localized token embedding $LET_{\mathbf{m}}^{(k)}$ to remain close to the original image embedding by minimizing the respective distance:

$$\mathcal{L}_{\text{reg}} = 1 - \cos\left(LET_{\mathbf{m}}^{(k)}, z_0^L\right). \tag{3}$$

The final loss function is a weighted sum of the Dice loss equation 2 and the regularization loss:

$$\mathcal{L} = \mathcal{L}_{\text{Dice}} + \alpha \cdot \mathcal{L}_{\text{reg}}, \tag{4}$$

where $\alpha \in \mathbb{R}$ is a hyperparameter modulating the influence of the regularization loss. It allows us to regulate how much regional vs. global information should be encoded in the output token embedding. We found that this helps for tasks that need context knowledge such as referring expressions retrieval(Wu et al., 2020), while 'object-only' tasks such as localized classification do not profit from such an alignment.

**Faster mask inversion via gradient decomposition** The derivation of the explainability map necessitates the calculation of a gradient, and similarly, each gradient descent iteration requires the

computation of a gradient with respect to the loss function $\mathcal{L}$. Consequently, this iterative process requires the evaluation of second-order derivatives of the form $\frac{\partial \mathcal{L}}{\partial LET_{\mathbf{m}}^{(k)}}(LET_{\mathbf{m}}^{(k)}, \nabla \mathbf{A})$.

These can be computationally intensive and numerically unstable. To enhance the computational efficiency of this process, it is advantageous to obviate the need for backpropagation to generate explainability maps at each iteration. We propose a gradient decomposition strategy that simplifies the gradient computation associated with the explainability method. For a given iteration $k$, the gradient decomposition can be expressed as follows:

$$\nabla \mathbf{A} = \frac{\partial s}{\partial \mathbf{A}} = \frac{\partial \bar{\mathbf{z}} \cdot \left(LET_{\mathbf{m}}^{(k)}\right)^T}{\partial \mathbf{A}} = \frac{\partial \bar{\mathbf{z}}}{\partial \mathbf{A}} \cdot \left(LET_{\mathbf{m}}^{(k)}\right)^T \in \mathbb{R}^{h \times n \times n} \tag{5}$$

where $h$ is the number of heads and $n$ is the number of visual tokens. This equation holds true because the mask $LET_{\mathbf{m}}^{(k)}$ is not dependent on the activations $\mathbf{A}^L$. By decomposing the gradient in this manner, the task of generating the explainability map transitions from a gradient computation to a dot product operation between $LET_{\mathbf{m}}^{(k)} \in \mathbb{R}^d$ and $\frac{\partial \bar{\mathbf{z}}}{\partial \mathbf{A}} \in \mathbb{R}^{h \times n \times n \times d}$. As a result, the proposed gradient decomposition approach significantly reduces the computational load by eliminating the need to compute the gradient of the score function $s$ with respect to the activations $\mathbf{A}$ multiple times. Instead, a single computation of the gradient $\frac{\partial \bar{\mathbf{z}}}{\partial \mathbf{A}}$ suffices for all subsequent gradient descent steps, thereby expediting the mask inversion process and enhancing its numerical stability.

## 4 EXPERIMENTS

### 4.1 DOWNSTREAM TASKS

We give a brief overview of the evaluated downstream tasks here. Please see section B for details. **Referring Expression Retrieval** To assess the proposed method's ability to capture localized properties, we first evaluate it for referring expression classification. Given an image and a set of masks, we generate an embedding for each mask and match the generated region embeddings to a set of text queries (referring expressions) encoded with the respective text encoder. The query mask whose localized embedding exhibits the highest cosine similarity with the text embedding of a referring expression is selected. We employ standard referring expression datasets: PhraseCut (Wu et al., 2020), RefCOCO, and RefCOCO+ (Kazemzadeh et al., 2014), reporting top-1, top-5, top-10 accuracy, mean Intersection over Union (mIoU) and overall Intersection over Union (oIoU).

**Class Retrieval** Zero-shot classification requires classifying an image by matching its visual embedding to the closes textual description of all classes present in the dataset. Here, we *classify a specific region* of the image, indicated by a query mask on an object, leveraging two semantic segmentation datasets, PascalVOC (Everingham et al., 2015) and PascalContext (Mottaghi et al., 2014), and one instance segmentation dataset, MSCOCO (Lin et al., 2014). The performance is evaluated using the top-1, top-5, and top-10 accuracy. Finally, we evaluate the proposed method in a large-scale open-vocabulary setting on a subset of the OpenImagesV7 (Benenson & Ferrari, 2022), which offers mask annotations for a diverse array of objects across 350 unique classes.

**Localized Captioning** Traditionally, image captioning models generate captions for entire images based on the visual representation provided by an image encoder. In contrast, we aim to evaluate MaskInversion's ability to focus the captioner on a specific region, while maintaining contextual relevance. To this end, we leverage a pretrained image captioner, CLIPCap (Mokady et al., 2021), and provide it with the localized embedding token of a query mask to generate a caption. CLIPCap is trained on top of the CLIP vision encoder and feeds its [CLS] token to GPT-2(Radford et al., 2019) to produce a caption. Here, we feed the localized embeddings of MaskInversion as a drop-in replacement of CLIP's [CLS] token to the captioner *without finetuning*. As no dataset directly supports this kind of evaluation, we adapted an existing dataset, PhraseCut(Wu et al., 2020). To quantitatively evaluate the generated localized captions, we match them to the set of ground-truth referring expressions for this image using the text encoder from CLIP (ViT-L/14 by OpenAI). We consider a caption as correct if its cosine similarity to the ground-truth referring expression for this mask is the highest among other referring expressions. The reported metric for this task is the top-1 accuracy.

| | Method | zero-shot | PhraseCut | | | RefCOCO | | | RefCOCO+ | | |
|---|---|---|---|---|---|---|---|---|---|---|---|
| | | | Acc@1 | Acc@5 | Acc@10 | Acc@1 | mIoU | oIoU | Acc@1 | mIoU | oIoU |
| CPT ‡ | RN50x16 + ViT-B/32 | ✓ | - | - | - | 32.2 | - | - | 31.9 | - | - |
| GradCAM‡ | RN50x16 + ViT-B/32 | ✓ | - | - | - | 42.9 | - | - | 47.8 | - | - |
| ReCLIP‡ | RN50x16 + ViT-B/32 | ✓ | - | - | - | 45.8 | - | - | 47.9 | - | - |
| RedCircle‡ | RN50x16 + ViT-L/14@336 | ✓ | - | - | - | 49.8 | - | - | 55.3 | - | - |
| FGVP‡ | RN50x16 + ViT-B/32 +ViT-L/14@336 | ✓ | - | - | - | 52.9 | - | - | 57.4 | - | - |
| AlphaCLIP ‡ | ViT-B/16+ViT-L/14 | ✗ | - | - | - | 55.7 | - | - | 55.6 | | - |
| RIS | ViT-B/32 | ✓ | - | - | - | - | - | 42.6 | - | - | 37.1 |
| CLIP* | ViT-B/16 | ✓ | 14.4 | 66.4 | 87.1 | 18.3 | 18.9 | 15.3 | 18.4 | 19.0 | 15.4 |
| Crop* | ViT-B/16 | ✓ | 15.1 | 67.0 | 87.6 | 17.9 | 18.5 | 15.5 | 19.0 | 19.5 | 16.1 |
| Masked Crop* | ViT-B/16 | ✓ | 48.3 | 89.7 | 97.2 | 52.3 | 52.9 | 41.2 | 58.7 | 59.4 | 47.5 |
| RedCircle* | ViT-B/16 | ✓ | 21.5 | 72.3 | 90.3 | 42.5 | 43.2 | 32.7 | 42.5 | 43.3 | 33.5 |
| FGVP* | ViT-B/16 | ✓ | 35.9 | 83.5 | 95.2 | 42.6 | 43.2 | 33.3 | 48.0 | 48.7 | 38.0 |
| AlphaCLIP* | ViT-B/16 | ✗ | 34.0 | 80.0 | 93.6 | 43.4 | 44.0 | 38.1 | 44.2 | 44.7 | 39.7 |
| MaskInversion | ViT-B/32 | ✓ | 54.8 | 93.0 | 98.5 | 54.1 | 54.7 | 42.3 | 55.8 | 56.5 | 44.3 |
| MaskInversion | ViT-B/16 | ✓ | 57.2 | 93.3 | 98.3 | 56.1 | 56.8 | 44.5 | 58.3 | 59.0 | 46.5 |
| MaskInversion | ViT-L/14 | ✓ | 60.2 | 94.9 | 98.7 | 56.1 | 56.7 | 42.0 | 60.2 | 60.9 | 47.5 |
| MaskInversion | ViT-H/14 | ✓ | **64.0** | **96.0** | **99.2** | **61.2** | **61.8** | **47.5** | **65.0** | **65.7** | **52.6** |

Table 1: Evaluation of MaskInversion on Referring Expression Retrieval. Given a query mask, the task is to retrieve the corresponding expression. ‡ indicates deviating evaluation settings where a pretrained region proposal is used and the prediction is counted as a hit if the matched region has an IoU> 0.5; in this setting, several proposals could result in a hit. *indicates reproduced results.

| | Method | PascalVOC | | | PascalContext | | | COCO | | | OpenImagesV7 | | |
|---|---|---|---|---|---|---|---|---|---|---|---|---|---|
| | | Acc@1 | Acc@5 | Acc@10 | Acc@1 | Acc@5 | Acc@10 | Acc@1 | Acc@5 | Acc@10 | Acc@1 | Acc@5 | Acc@10 |
| ViT-B/16 | CLIP* | 40.1 | 87.2 | 95.6 | 17.8 | 38.7 | 52.7 | 25.0 | 54.9 | 72.6 | 28.9 | 63.4 | 72.7 |
| | Crop* | 27.9 | 51.2 | 72.4 | 5.6 | 13.2 | 20.4 | 23.9 | 34.5 | 41.5 | 0.8 | 3.8 | 7.05 |
| | Masked Crop* | 75.0 | 91.4 | 96.4 | 40.4 | 65.9 | 75.8 | 38.2 | 57.7 | 65.2 | 33.8 | 61.9 | 73.7 |
| | RedCircle* | 47.5 | 92.9 | 97.7 | 21.3 | 45.0 | 57.4 | 28.8 | 63.0 | 77.3 | 40.5 | 75.8 | 84.5 |
| | AlphaCLIP* | 52.6 | 85.9 | 93.8 | 27.7 | 60.9 | 75.1 | 30.9 | 55.9 | 70.3 | 43.0 | 77.4 | 84.3 |
| | FGVP* | 71.8 | 93.6 | 98.3 | 32.6 | 58.9 | 72.4 | 35.9 | 62.2 | 72.6 | 39.4 | 75.6 | 84.6 |
| | RIS* | 78.0 | 95.2 | 98.1 | 38.1 | 62.7 | 74.3 | 43.6 | 65.3 | 72.4 | 34.5 | 66.5 | 75.8 |
| B/32 | MaskInversion | 79.5 | 96.4 | 98.8 | 46.7 | 74.9 | 84.6 | 38.0 | 65.8 | 78.4 | 42.6 | 78.8 | 86.6 |
| B/16 | MaskInversion | 85.4 | 96.4 | 98.8 | 58.1 | 83.7 | 90.5 | 44.7 | 71.6 | 83.0 | 46.3 | 80.4 | 87.9 |
| L/14 | MaskInversion | 91.0 | 99.1 | 99.8 | 59.0 | 86.3 | 92.5 | 56.0 | 84.2 | 91.4 | 48.7 | 81.0 | 88.1 |
| H/14 | MaskInversion | **93.5** | **99.4** | **99.7** | **61.8** | **86.0** | **91.8** | **63.7** | **88.3** | **93.5** | **51.2** | **85.2** | **91.4** |

Table 2: Comparison with baselines on Class Retrieval for Segmentation Datasets. Given a mask, the task is to retrieve the corresponding class. ∗ indicates reproduced results.

## 4.2 SETUP

The proposed method is evaluated using pretrained CLIP vision-language models. For ViT-B/32, ViT-B/16, and ViT-L/14, we used the original weights from OpenAI (Radford et al., 2021), and for ViT-H/14, we used the weights "laion2b_s32b_b79k" from the OpenCLIP library (Cherti et al., 2023; Schuhmann et al., 2022). For the MaskInversion optimization, we use the AdamW optimizer(Kingma, 2014) with 10 gradient descent iterations. We set the regularization parameter $\alpha$ to 5 for RefCOCO and RefCOCO+, and to 0 for all other datasets.

## 4.3 COMPARISON TO THE STATE-OF-THE-ART (SOTA)

**Referring Expression Retrieval** Table 1 presents the results on referring expression datasets. For some related approaches (CPT, GradCAM, ReCLIP, FGVP and RedCircle), as there is no directly comparable setting, we provide both the results as reported in their paper, and our reproduced results. Note that the original evaluation settings can vary for different methods. For reproduced results indicated by * we adapt the evaluation setting to the case where ground-truth masks are used as described in Sec. 4.1. We used the code provided by the authors of each method, encode each image together with the ground-truth masks of MSCOCO, and match the resulting representation to the text embedding produced by the respective backbone. We further compare with the following baselines: *CLIP* refers to the general CLIP baseline by using the image CLS token, *Crop* uses the CLS token of cropped region by forwarding only this region through CLIP, and *Masked Crop* refers to forwarding the full image, but keeping only the masked region and replacing all other pixels with the average pixel value of the dataset. On the PhraseCut dataset, MaskInversion outperforms all baselines, regardless of the model size. On the RefCOCO and RefCOCO+ datasets, MaskInversion also achieves SOTA performance. In addition, MaskInversion's performance scales well with the backbone size, establishing a new SOTA on every dataset when ViT-H/14 is used.

| Method | Backbone | VOC | Context | COCO | PhraseCut | RefCOCO | RefCOCO+ | OpenImagesV7 | Avg |
|--------|----------|-----|---------|------|-----------|---------|----------|--------------|-----|
| MaskCLIP | B/16 | 74.9 | 43.0 | 40.2 | 53.9 | 49.3 | 52.6 | 45.6 | 51.4 |
| CLIPSurgery | B/16 | 70.8 | 53.5 | 41.7 | 52.5 | 48.9 | 52.0 | **49.5** | 52.7 |
| SCLIP | B/16 | 64.3 | 43.0 | 33.4 | 37.2 | 40.7 | 42.4 | 45.5 | 43.8 |
| **Ours** | B/16 | **85.4** | **58.1** | **44.7** | **57.2** | **56.1** | **58.3** | 46.3 | **58.0** |
| MaskCLIP | L/14 | 55.1 | 33.2 | 29.3 | 47.6 | 43.2 | 47.2 | 32.5 | 41.2 |
| CLIPSurgery | L/14 | 78.3 | 46.4 | 47.7 | 47.2 | 47.3 | 50.9 | 45.5 | 51.9 |
| SCLIP | L/14 | 43.0 | 24.9 | 25.9 | 19.0 | 32.8 | 32.5 | 38.3 | 30.9 |
| **Ours** | L/14 | **91.0** | **59.0** | **56.0** | **60.2** | **56.1** | **60.2** | **48.7** | **61.6** |
| MaskCLIP | H/14 | 61.8 | 37.8 | 30.9 | 45.9 | 34.6 | 39.6 | 36.9 | 41.1 |
| CLIPSurgery | H/14 | 68.0 | 40.8 | 40.1 | 41.5 | 43.2 | 46.7 | 45.8 | 46.6 |
| SCLIP | H/14 | 38.2 | 20.7 | 19.8 | 15.2 | 20.7 | 20.7 | 35.6 | 24.4 |
| **Ours** | H/14 | **93.5** | **61.8** | **63.7** | **64.0** | **61.2** | **65.0** | **51.2** | **65.0** |

Table 3: Performance comparison of MaskInversion against different training-free method for different tasks. The reported numbers are Accuracies (higher is better).

| Mask Type | Acc |
|-----------|-----|
| Mask | 44.7 |
| Erosion | 42.7 |
| Dilation | 44.3 |
| Box | 42.9 |
| Box + SAM | **45.0** |

Table 4: **Mask Quality Ablation:** Class retrieval accuracy on MSCOCO for mask variations, as well as for bounding boxes and SAM masks based on bounding boxes.

| #Mask | Decomp. | Sec.↓ |
|-------|---------|-------|
| 5 | ✗ | **0.10** |
| 5 | ✓ | 0.13 |
| 10 | ✗ | 0.15 |
| 10 | ✓ | **0.14** |
| 50 | ✗ | 0.65 |
| 50 | ✓ | **0.27** |
| 100 | ✗ | 1.27 |
| 100 | ✓ | **0.44** |

Table 5: **Gradient Decomposition:** Runtime with and w/o gradient decomposition for different number of masks.

| Method | Acc |
|--------|-----|
| CLIP | 20.1 |
| AlphaCLIP | 31.8 |
| MaskInversion | **48.4** |

Table 6: **Localized captioning Ablation:** We use CLIP-Cap to generate captions based on embeddings generated by CLIP, AlphaCLIP, and Mask-Inversion corresponding to the region highlighted by the mask. We report top-1 image-to-text retrieval accuracy.

**Class Retrieval** Table 2 compares MaskInversion to other methods for the case of zero-shot class retrieval, keeping the same setting as for Referring Expression Retrieval. MaskInversion outperforms all other methods on all datasets on semantic segmentation datasets, such as PascalVOC and Pascal-Context. Furthermore, MaskInversion also exhibits good performance on the instance segmentation dataset COCO. These results demonstrate that MaskInversion can effectively direct the attention of the foundation model to *multiple instances of the same object class at the same time, as well as to a single instance*. Here, MaskInversion also outperforms the recently proposed AlphaCLIP (Sun et al., 2024), which fine-tunes CLIP on millions of mask-text pairs annotations, thereby demonstrating its ability to excel without the need to fine-tune CLIP. Finally, on OpenImagesV7, which features a much larger vocabulary of 350 classes, we can see that methods like AlphaCLIP perform well, as they are specifically trained for such tasks. Nonetheless, MaskInversion again outperforms all other methods we compared, demonstrating its capability to handle large vocabularies.

**Comparison with Training-free Methods** We compare MaskInversion with recent training-free approaches for mask-conditioned CLIP representations: MaskCLIP Zhou et al. (2022), CLIPSurgery Li et al. (2023), and SCLIP Wang et al. (2024). These methods compute patch token representations and aggregate tokens within the mask region through average pooling to obtain localized embeddings. Using their official implementations, we evaluate these approaches across our benchmark suite while maintaining our evaluation protocol. Table 3 presents the comparative results across different CLIP architectures (ViT-B/16, ViT-L/14, and ViT-H/14). MaskInversion consistently outperforms training-free methods across nearly all datasets and model scales. The performance gap becomes particularly pronounced with larger backbone architectures, where MaskInversion demonstrates substantial improvements. For instance, with ViT-H/14, our method achieves gains of +31.7%, +21.0%, and +32.8% on VOC, Context, and COCO datasets respectively compared to the best performing baseline. Notably, while training-free methods show competitive performance with ViT-B/16, their effectiveness diminishes with larger architectures. In contrast, MaskInversion effectively leverages the increased capacity of larger models, showing consistent performance improvements across all scales. This suggests that our approach better utilizes the representational power of advanced CLIP models while maintaining robust performance across different architectural scales.

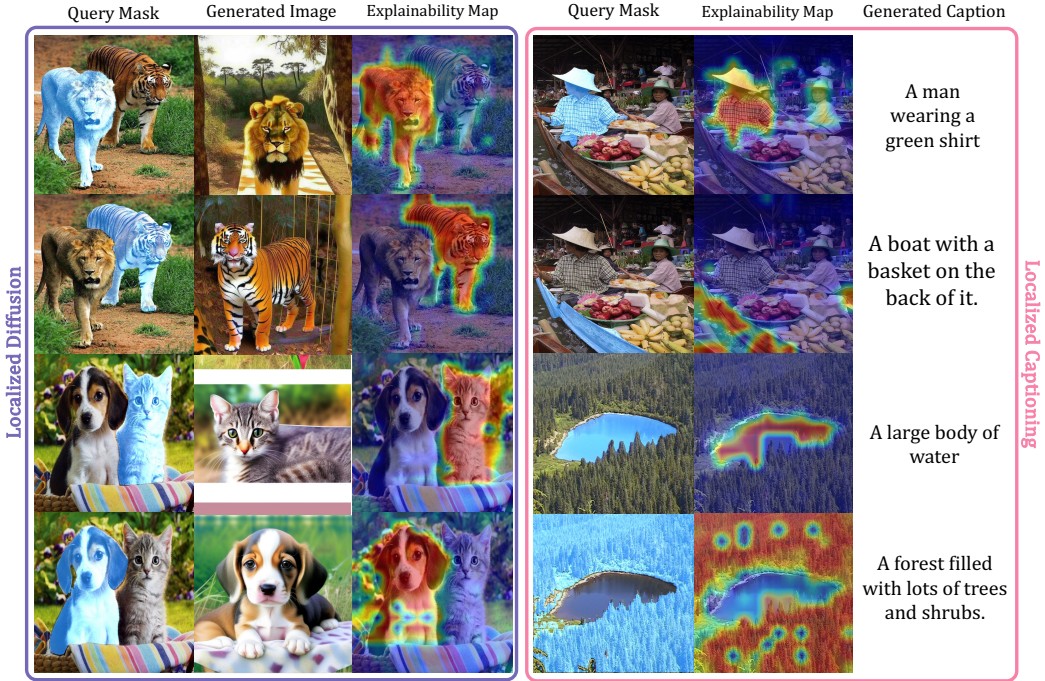

Figure 3: **Localized Embedding Visualizations:** Visualisation of the learned localized embedding using *(left)* a pretrained diffusion model; *(right)* an image captioner. In both cases, the global feature representation is replaced by the output of MaskInversion depending on the query mask.

## 4.4 ABLATIONS

**Impact of Mask Quality** MaskInversion utilizes an input query mask to direct the output of the foundation model toward the area covered by the mask. Given that the mask is a critical element of the MaskInversion method, we explore here how variations in mask quality affect its performance. To this end, we evaluate different mask conditions for the task of Class Retrieval on the MSCOCO dataset as shown in Table 4 and Figure 9: *Box* uses the masks' bounding-boxes instead of precise segmentation masks, *Box+SAM* feeds the bounding-boxes to SAM (Kirillov et al., 2023) to produce approximate masks and uses those instead of ground-truth masks, and *Erosion* and *Dilation* apply the respective morphological operations to the original masks. The results indicate that eroding the mask leads to a more substantial decrease in performance compared to dilation. We further see a decrease in accuracy from $44.7\%$ to $42.9\%$ when using bounding-boxes only, whereas the combination of bounding-boxes and SAM to derive the mask achieves comparable performance to inputting ground-truth mask. This scenario is especially relevant for practical applications where users may find it easier to draw bounding-boxes rather than detailed masks.

**Runtime Evaluation for Gradient Decomposition** Table 5 presents a runtime comparison of the vanilla MaskInversion, where the gradient gradient-based explainability map is computed at each iteration for each mask, versus the "gradient-decomposition" proposed in section 3.2 for $K = 10$ steps. When there are more than 5 masks in an image, the proposed gradient decomposition is faster than the vanilla way of computing the explainability map (see appendix Sec. I for an ablation on the number of iterations).

## 4.5 LOCALIZED CAPTIONING ANALYSIS

We further consider the performance of MaskInversion against CLIP and AlphaCLIP for localized captioning in Table 6. We start from CLIPCap (Mokady et al., 2021) as the base captioner and replace the CLIP image encoder with either AlphaCLIP or the output of MaskInversion without any fine-tuning. We observe that MaskInversion demonstrates the ability to focus the captioner on the region of interest, as the accuracy more than doubles when using MaskInversion versus only using CLIP. Moreover, MaskInversion also significantly outperforms AlphaCLIP, despite not involving any fine-tuning of the CLIP model. Figure 3 presents qualitative examples of the localized captions generated by MaskInversion+CLIPCap for different query masks. MaskInversion demonstrates a

high degree of precision in focusing the captioning model on specific image regions specified by the query masks. Both, the caption and the heatmap, focus on the area covered by the query mask.

## 4.6 Mask Embedding for Image Diffusion

To further visualize the concepts captured in the learned representation output by MaskInversion, we employed $\lambda$-ECLIPSE (Patel et al., 2024), which takes as input a visual embedding from a ViT-bigG/14 CLIP model along with a text prompt and generates variations of the input image that correspond to the prompt. Utilizing the default settings of $\lambda$-ECLIPSE as described in (Patel et al., 2024), we generate images based on different query masks used within the MaskInversion process. As Figure 3 show, resulting images vary depending on the mask used. The images focus on the objects inside the query mask, confirming that MaskInversion directs the model's attention to specific parts of the image. Moreover, we observe that the final explainability map is focused on the area covered by the mask, validating the effectiveness of our proposed optimization process.

## 5 Conclusion

We proposed MaskInversion as a method to create region embeddings that are grounded in the rich feature representations of foundation models, without the need to fine-tune the model. To this end, we leveraged explainability maps to learn an embedding vector focused on a specific image region. We extend this idea with an add-on regularization loss to balance global and local representations, and with a gradient decomposition technique to improve runtime.

## Acknowledgment

Walid Bousselham is supported by the German Federal Ministry of Education and Research (BMBF) project STCL - 01IS22067. The authors gratefully acknowledge the Gauss Centre for Supercomputing e.V. (www.gauss-centre.eu) for funding this project by providing computing time on the GCS Supercomputer JUPITER — JUWELSJülich Supercomputing Centre (2021) at Jülich Supercomputing Centre (JSC). This work also acknowledges support from the project *ELLIOT-FM: Open Multi-Modal Foundation Models with Strong Generalization and Reasoning*.

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

# APPENDIX

## A  OVERVIEW

In the Appendix, we first provide additional details on the different downstream tasks in Sec.B. Sec.C provides a visualization of the explainability map throughout the optimization process. Sec.4.3 presents a comprehensive comparison with training-free methods. Sec.F analyzes the influence of the hyperparameter $\alpha$ on balancing local and global information. Sec.G demonstrates our method's capability to handle multiple objects. Sec.H provides visualizations of the mask distortion used for our ablations. Sec.I presents an ablation of the proposed gradient decomposition technique. Sec.J evaluates the impact of different explainability methods on MaskInversion. Sec.K and Sec.L respectively discuss the limitations of SOTA methods and the proposed MaskInversion. Finally, we provide additional qualitative examples of localized captioning and diffusion in Sec.M and Sec.N.

| Query Mask | Step1 | Step 3 | Step 5 | Step 7 | Step 10 |

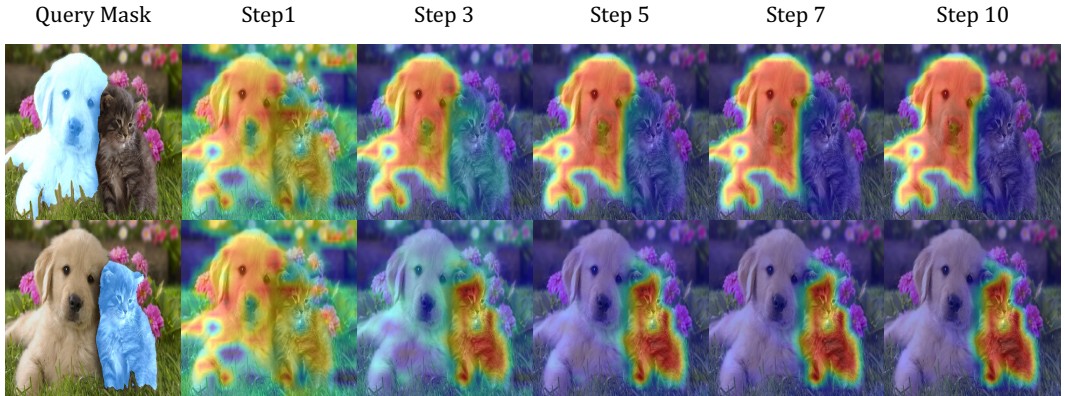

Figure 4: **Visualization of the Explainability Maps throughout the optimization steps.**

## B    DOWNSTREAM TASKS

**Referring Expressions**    To assess the proposed method's ability to capture localized properties, we evaluate it for referring expression classification. Given an image and a set of masks, we generate an embedding for each mask within an image and match the generated region embeddings to a set of text queries (referring expressions) encoded with the respective text encoder. The query mask whose localized embedding exhibits the highest cosine similarity with the text embedding is selected. We employ standard referring expression datasets, i.e. PhraseCut (Wu et al., 2020), RefCOCO, and RefCOCO+ (Kazemzadeh et al., 2014). For RefCOCO and RefCOCO+, we use the mask annotations from the MSCOCO (Lin et al., 2014) dataset, which has about 30 masks per image, thereby increasing the difficulty of the task. For PhraseCut, we consider the masks of all annotated referring expressions as candidates, reporting top-1, top-5, and top-10 accuracy. Additionally, following (Subramanian et al., 2022; Sun et al., 2024; Yang et al., 2023; Shtedritski et al., 2023), for RefCOCO and RefCOCO+, we report the mean Intersection over Union (mIoU) and overall Intersection over Union (oIoU).

**Class Retrieval**    Second, we consider the task of zero-shot classification as a common benchmark for vision-language models. In that task, an image is classified by matching its visual embedding with the textual description of the classes present in the dataset. Here, we propose to increase the granularity by using it to *classify a specific region* of the image: given a query mask of an object, classify it by matching its localized embedding to the text embeddings of the classes in the datasets. For this, we leverage two semantic segmentation datasets, PascalVOC (Everingham et al., 2015) and PascalContext (Mottaghi et al., 2014), with 19 and 59 classes, respectively, and one instance segmentation dataset, MSCOCO (Lin et al., 2014), with 80 classes. The performance is evaluated using the top-1, top-5, and top-10 accuracy metrics, denoted by $Acc@1$, $Acc@5$, and $Acc@10$. Finally, we challenge the proposed method in a large-scale open-vocabulary setting by using a dataset encompassing a substantially larger number of classes. We utilize a subset of the OpenImagesV7 (Benenson & Ferrari, 2022) dataset, which offers mask annotations for a diverse array of objects across 350 unique classes. The evaluation metrics are again top-1, top-5, and top-10 accuracy reported as $Acc@1$, $Acc@5$, and $Acc@10$.

**Localized Captioning**    Traditionally, image captioning models generate captions for entire images based on the visual representation provided by an image encoder. In contrast, we aim to evaluate our method's ability to focus the captioner on a specific image region while maintaining contextual relevance. To this end, we leverage a pretrained image captioner, CLIPCap (Mokady et al., 2021), and provide it with the localized embedding token of a query mask to generate a caption. CLIPCap is trained on top of the CLIP vision encoder and feeds its [CLS] token to GPT-2(Radford et al., 2019) to produce a caption. Here, we feed the localized embeddings of MaskInversion as a drop-in replacement of the CLIP [CLS] token to the captioner ***without any finetuning***. As no dataset directly supports this evaluation type, we adapted an existing dataset, PhraseCut. To quantitatively evaluate the generated localized captions, we match the generated caption to the set of ground truth referring expressions for this image using the text encoder from CLIP (ViT-L/14 by OpenAI), consider the

caption correct if the cosine similarity between the generated caption and the ground truth referring expression for this mask is the highest. The reported metric for this task is the top-1 accuracy.

## C  OPTIMIZATION STEPS VISUALIZATION

Finally, Figure 4 provides a visualization of the explainability map throughout the optimization process employed by MaskInversion. It is observed that the explainability map increasingly concentrates on the region covered by the query mask as the optimization progresses. This observation is indicative of the method's ability to effectively focus the attention of the underlying foundation model on the designated areas of the image.

## D  CONVERGENCE ANALYSIS AND HYPERPARAMETER CHOICE

In Section 3.2, we set the number of gradient descent iterations for the optimization of the localized embedding token at $K = 10$. This value was determined through an empirical analysis designed to balance the quality of the learned embedding with computational efficiency.

To illustrate this choice, we monitor the optimization process on the PascalVOC dataset. Figure 5 presents the evolution of both the optimization loss (left y-axis, red curve) and the downstream classification accuracy (right y-axis, blue curve) over 100 iterations.

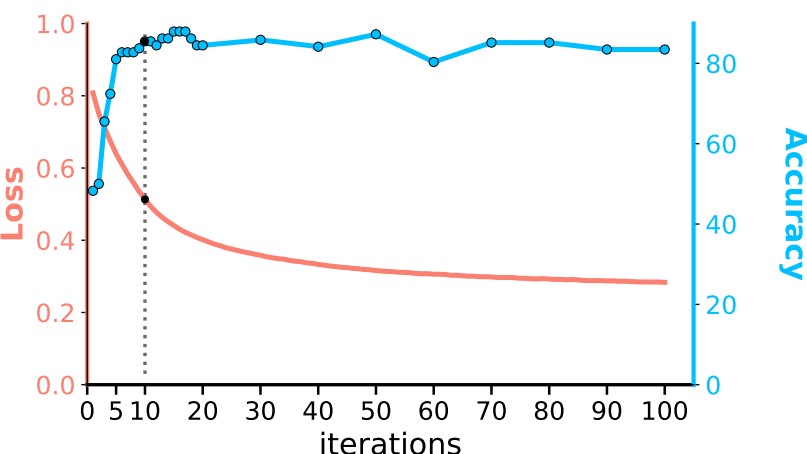

Figure 5: **Convergence Analysis of MaskInversion.** The plot illustrates the optimization loss (red, left axis) and the resulting accuracy on PascalVOC (blue, right axis) over iterations. The dotted line marks the chosen stopping point at $K = 10$ iterations.

As demonstrated in Figure 5, the optimization process converges quickly. The classification accuracy increases sharply in the initial steps, reaching approximately $85\%$ by iteration 10, after which it effectively plateaus. This asymptotic behavior indicates that the localized embedding token $LET_{\mathbf{m}}$ quickly captures the core semantic features required for the task, with the explainability map becoming sufficiently aligned with the query mask within the first 10 iterations.

While the loss function continues to decrease marginally beyond the $10^{\text{th}}$ iteration, these residual improvements primarily correspond to boundary refinements between the generated explainability map and the binary query mask. As evidenced by our ablation experiments on mask quality (see Table 4 in the main text), such minor boundary adjustments have a minimal impact on the semantic quality of the learned embedding. Consequently, extending the optimization beyond this point incurs additional computational cost without yielding significant performance gains. Therefore, we select $K = 10$ to ensure high-quality localized embeddings while maintaining computational efficiency.

# E    EMPIRICAL RUNTIME ANALYSIS

A key consideration for high-resolution tasks, such as medical imaging or aerial photography, is how the computational cost of this method scales with increased input resolution. To empirically validate this scalability, we measured the inference runtime while varying the input image resolution. We tested standard resolutions of $224 \times 224$ pixels and increased up to $768 \times 768$ pixels. We a patch size of $16 \times 16$, this corresponds to a range of $n = 196$ to $n = 2,304$ visual tokens. The experiment was conducted with a fixed budget of $K = 10$ iterations processing 10 masks per image.

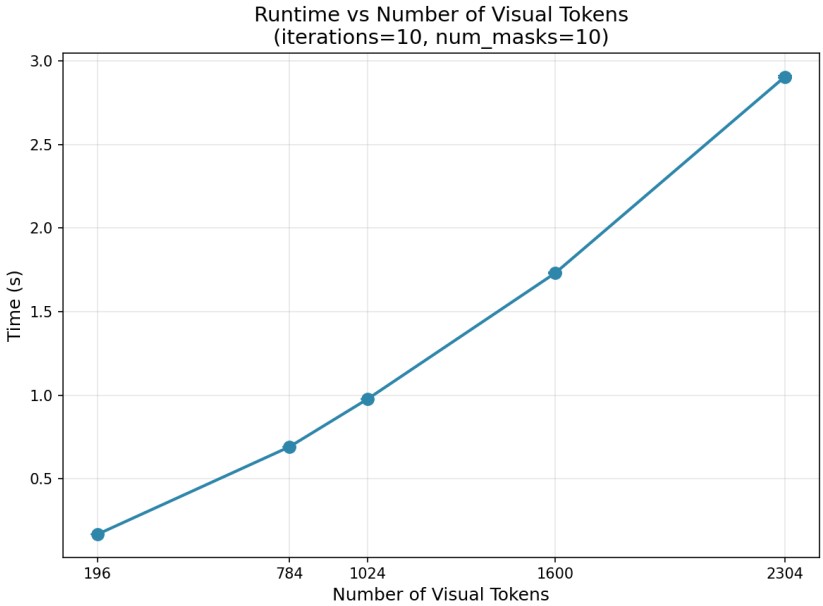

Figure 6: **Runtime Scalability.** The plot displays the processing time (in seconds) as a function of the number of visual tokens $n$. The input resolutions correspond to $224^2$, $448^2$, $512^2$, $640^2$, and $768^2$. The results demonstrate a clear linear relationship between runtime and the number of tokens.

The results are illustrated in Figure 6. As predicted, the runtime shows a linear scaling with the number of visual tokens. For instance, increasing the resolution such that the token count doubles results in an approximate doubling of the processing time. This linear scaling confirms that *MaskInversion* remains computationally tractable even for higher-resolution inputs.

# F    INFLUENCE OF $\alpha$

| alpha | 0.0 | 0.5 | 1.0 | 1.5 | 2.0 | 2.5 | 3.0 | 4.0 | 5.0 | 5.5 | 6.0 | 6.5 | 7.0 | 7.5 | 8.0 | 10.0 | 20.0 |
|---|---|---|---|---|---|---|---|---|---|---|---|---|---|---|---|---|---|
| Acc | 41.7 | 47.6 | 50.3 | 52.2 | 53.9 | 54.6 | 55.2 | 56.0 | 56.2 | 56.0 | 55.8 | 56.0 | 56.2 | 56.1 | 55.8 | 53.7 | 20.5 |

Table 7: Accuracy for different values of $\alpha$ on RefCOCO.

We conduct an extensive analysis of the hyperparameter $\alpha$ to understand its role in balancing local and global information within the learned embeddings. Figure 7 illustrates this effect through generated captions for different $\alpha$ values. When $\alpha = 0$, the model generates descriptions focused strictly on the masked region (*e.g.*, "woman in a boat"), while increasing $\alpha$ progressively incorporates more contextual information(*e.g.*, "produce" or "vegetables"). Quantitatively, we observe that performance on RefCOCO improves as $\alpha$ increases from 0 (41.7%) to an optimal value around $\alpha = 5.0$ (56.2%), before gradually declining for larger values. This sweet spot ($\alpha \approx 5.0$) represents an optimal balance where the embedding retains sufficient local information while leveraging beneficial contextual cues. Beyond $\alpha > 7.5$, performance deteriorates as the representation becomes increasingly similar to the global [CLS] token, with a dramatic drop at $\alpha = 20.0$ (20.5%). This analysis demonstrates that $\alpha$

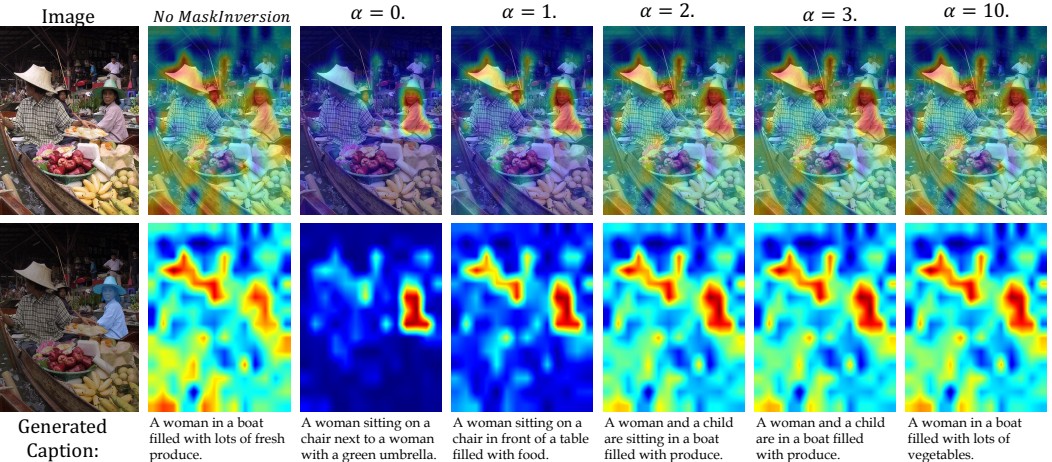

Figure 7: Qualitative analysis of the influence of $\alpha$ on the generated captions.

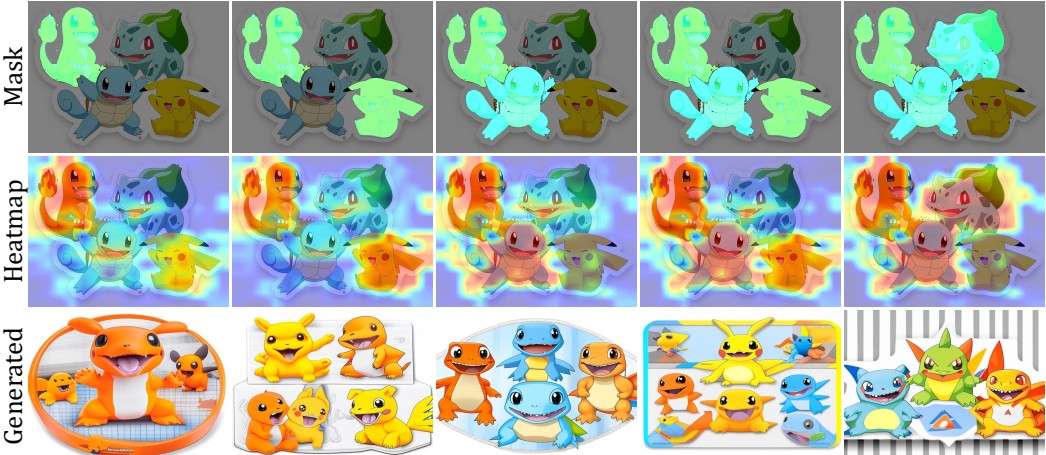

Figure 8: **Multi-Object Analysis:** Visualization of MaskInversion's ability to handle multiple objects. *(top)* Query masks highlighting different combinations of objects, *(middle)* corresponding heatmaps showing the model's focus regions, and *(bottom)* generated images using $\lambda$-ECLIPSE demonstrating the preservation of multiple object characteristics in the learned embeddings.

effectively functions as a control mechanism for trading off local detail against global context in the learned representations.

# G    MULTI-OBJECT

While quantitative evaluation of multi-object scenarios presents inherent challenges, we demonstrate MaskInversion's capability to handle multiple objects through qualitative analysis. As shown in Figure 8, our method effectively captures the relationships and context of multiple objects within a single mask. For instance, when given a mask covering multiple Pokémon characters, the generated diffusion outputs maintain coherent representations of all objects while preserving their spatial relationships and individual characteristics. The diffusion model successfully reconstructs multiple objects from the localized embedding, indicating that MaskInversion effectively encodes information about multiple entities and their relative positioning. This is particularly evident in cases where the mask encompasses groups of similar objects (e.g., multiple Pokémon) or diverse object combinations, demonstrating the method's robustness in handling complex, multi-object scenarios without losing individual object details or their contextual relationships .

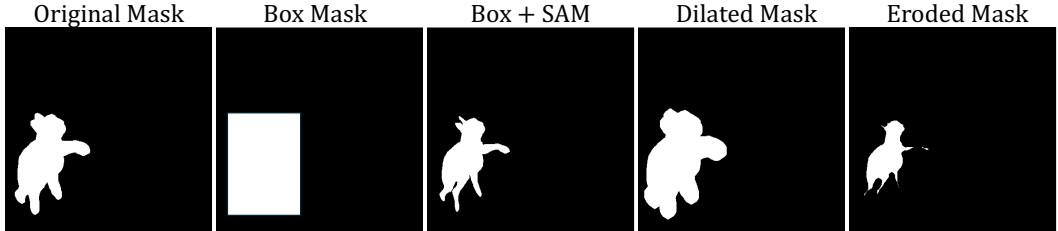

Figure 9: **Mask Quality Ablation:** example of different mask degradation settings.

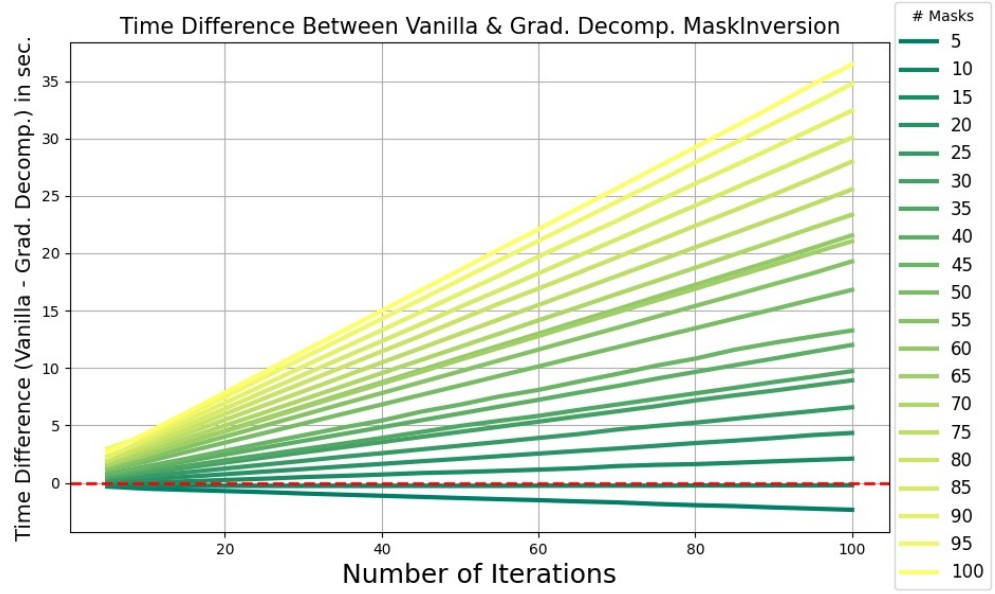

Figure 10: **Gradient Decomposition:** Time difference between using or not using the gradient decomposition technique described Sec.3.2, using ViT-B/16 for different numbers of masks and iterations ranging from 5 to 100. The time difference is in seconds.

## H    MASK QUALITY

Figure 9 provides a visualization of the different mask degradation settings entertained in Table 4.

## I    GRADIENT DECOMPOSITION

Figure 10 provides a more thorough comparison of the vanilla MaskInversion process described in Section 3.2 against the gradient decomposition trick described in Section 3.2. Namely, Figure 10 extends Table 5 to different numbers of gradient descent iterations and to more number of masks.

### I.1    DETAILED DERIVATION OF GRADIENT DECOMPOSITION

In Section 3.2, we introduced a gradient decomposition strategy to enhance the computational efficiency of the *MaskInversion* process. This strategy relies on the independence of the localized embedding token $LET_{\mathbf{m}}^{(k)}$ from the model activations $\mathbf{A}$ during the gradient computation step. Here, we provide a more detailed formal proof supporting Equation 5.

**Problem Formulation** Let the scalar activation score $s$ be defined as the dot product between the global image representation $\bar{\mathbf{z}}$ and the localized embedding token $LET_{\mathbf{m}}^{(k)}$ at iteration $k$. For clarity, we omit normalization terms (as in cosine similarity), noting that this does not alter the dependency logic:

$$s = \bar{\mathbf{z}} \cdot (LET_{\mathbf{m}}^{(k)})^T, \tag{6}$$

where:

- $\bar{\mathbf{z}} = \frac{1}{n} \sum_p z_p := f(\mathbf{A})$ represents the combined patch and [CLS] token representation averaged across spatial dimensions. This term is a direct function of the intermediate activations $\mathbf{A}$.
- $LET_{\mathbf{m}}^{(k)}$ represents the learnable vector parameters at the current optimization step $k$.

**Derivation** Our objective is to compute the gradient of the score $s$ with respect to the activations $\mathbf{A}$, denoted as $\nabla \mathbf{A}$. By applying the product rule of calculus, the gradient can be expanded as follows:

$$\nabla \mathbf{A} = \frac{\partial s}{\partial \mathbf{A}} = \frac{\partial \left( \bar{\mathbf{z}} \cdot (LET_{\mathbf{m}}^{(k)})^T \right)}{\partial \mathbf{A}} \tag{7}$$

Expanding the terms yields:

$$\frac{\partial s}{\partial \mathbf{A}} = \left( \frac{\partial \bar{\mathbf{z}}}{\partial \mathbf{A}} \cdot (LET_{\mathbf{m}}^{(k)})^T \right) + \left( \bar{\mathbf{z}} \cdot \frac{\partial (LET_{\mathbf{m}}^{(k)})^T}{\partial \mathbf{A}} \right). \tag{8}$$

**Proof of Independence** To evaluate the second term in Equation equation 8, we examine the definition of the localized token. While $LET_{\mathbf{m}}^{(0)}$ may be initialized using the global [CLS] token (which is derived from $\mathbf{A}$), in our implementation, the token is strictly **detached** from the computational graph upon initialization:

$$LET_{\mathbf{m}}^{(0)} := \text{stop\_gradient}([\text{CLS}]). \tag{9}$$

For all subsequent iterations $k$, $LET_{\mathbf{m}}^{(k)}$ is treated as an external, standalone optimization variable updated via the optimizer, rather than a continuous function of the image input from the current forward pass. Consequently, the partial derivative of the token with respect to the current activations is zero:

$$\frac{\partial LET_{\mathbf{m}}^{(k)}}{\partial \mathbf{A}} = \mathbf{0}. \tag{10}$$

Substituting this into Equation equation 8, the second term vanishes:

$$\begin{aligned} \nabla \mathbf{A} &= \left( \frac{\partial \bar{\mathbf{z}}}{\partial \mathbf{A}} \cdot (LET_{\mathbf{m}}^{(k)})^T \right) + (\bar{\mathbf{z}} \cdot \mathbf{0}) \\ &= \frac{\partial \bar{\mathbf{z}}}{\partial \mathbf{A}} \cdot (LET_{\mathbf{m}}^{(k)})^T. \end{aligned} \tag{11}$$

This derivation confirms that the decomposition presented in Equation (5) is mathematically exact. It allows for the pre-computation of the Jacobian $\frac{\partial \bar{\mathbf{z}}}{\partial \mathbf{A}}$, which can then be reused across all $K$ iterations via a simple dot product with the evolving token $LET_{\mathbf{m}}^{(k)}$, reducing computational cost.

## J   IMPACT OF THE EXPLAINABILITY METHOD

Given that MaskInversion leverages an explainability method to guide the inversion process, its dependency on the choice of explainability method was evaluated. We experimented with alternative gradient-based methods, such as GradCAM and CheferCAM, in place of the originally used LeGrad. The comparative results on the MSCOCO dataset are presented in Table 8. LeGrad significantly outperformed the other methods, which can be attributed to its design specificity for ViT architectures, unlike GradCAM and CheferCAM, which are tailored for CNNs and general transformers,

| Expl. Method | Acc@1 |
|---|---|
| GradCAM | 34.6 |
| GradCAM[‡] | 47.6 |
| CheferCAM | 12.6 |
| LeGrad | 85.4 |

Table 8: **Explanability Method Ablation:** MaskInversion performance using different explainability methods on the class retrieval task on PascalVOC. ‡indicates a modified version of GradCAM without the ReLU operation.

| Method | Finetune Model | Modify Img. | Description |
|---|---|---|---|
| Crop | ✗ | ✓ | Crop the input image, thus losing the context |
| RedCircle | ✗ | ✓ | Draw a red circle around the area of interest. Contingent on the biases in the training data and modifying the image can cause a domain gap. |
| Masked Crop | ✗ | ✓ | Crop the input image and mask the background. |
| FGVP(Yang et al., 2023) | ✗ | ✓ | Heavily blur the background, thus losing the context. |
| RIS(Yu et al., 2023) | ✗ | ✓ | Masks the features of the ViT after a certain number of layers to prevent the [CLS] token to aggregate information from outside the mask. |
| AlphaCLIP(Sun et al., 2024) | ✓ | ✗ | Finetunes CLIP to take as input an image and a mask. AlphaCLIP was trained on fine-grained mask/text pairs. |

Table 9: On one hand, directly modifying the input pixels can cause a domain gap between what the model was trained on and what it is used for (e.g., RedCircle & Masked Crop). Moreover, it can also completely remove the context that can be crucial for downstream tasks (e.g., Crop & Masking). On the other hand, finetuning the model can not only result in forgetting the knowledge accumulated during pretraining but also requires fine-grained mask/text data (*e.g.* AlphaCLIP). Also, the training needs to be done for every model.

respectively. This finding aligns with the observations in (Bousselham et al., 2025), where LeGrad demonstrated superior localization capabilities essential for the tasks addressed by MaskInversion. Thus, the selection of an appropriate explainability method is crucial for optimizing the performance of MaskInversion.

## K    SOTA METHODS' LIMITATIONS

Table 9 provides a description of the different baselines we compare MaskInversion to.

## L    LIMITATIONS

Firstly, the efficacy of MaskInversion is inherently tied to the availability and quality of explainability methods that integrate well with the foundation model used. Models lacking robust explainability frameworks may not fully benefit from the MaskInversion approach, as the method relies on accurate and interpretable explanations to guide the inversion process. Consequently, the performance of MaskInversion may degrade when applied to models with suboptimal explainability methods.

Secondly, foundational models like CLIP are often trained on using small-resolution images, usually $224 \times 224$. This characteristic imposes a downstream limitation on the MaskInversion method, particularly when the task involves focusing the model's attention on small objects within the image. The reduced resolution can hinder the method's ability to accurately capture fine-grained details, thereby affecting the overall performance in scenarios requiring high precision on small-scale features. To mitigate that problem, in this work, we used bicubic interpolation on the pretrained positional embedding of the ViT to increase the resolution at inference from $224 \times 224$ to $448 \times 448$.

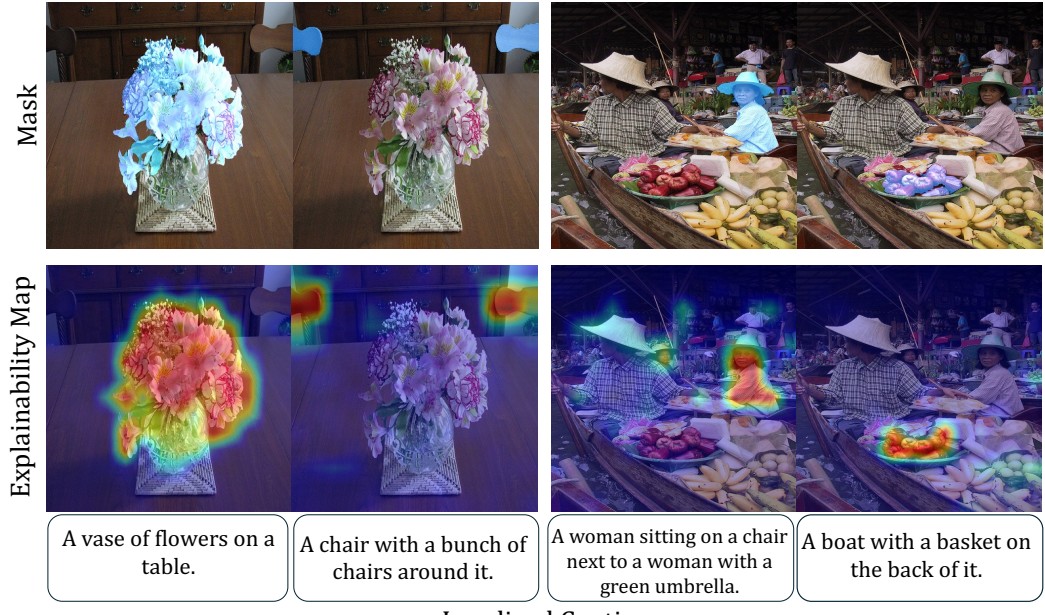

Localized Caption

Figure 11: **Additional Localized Captions.**

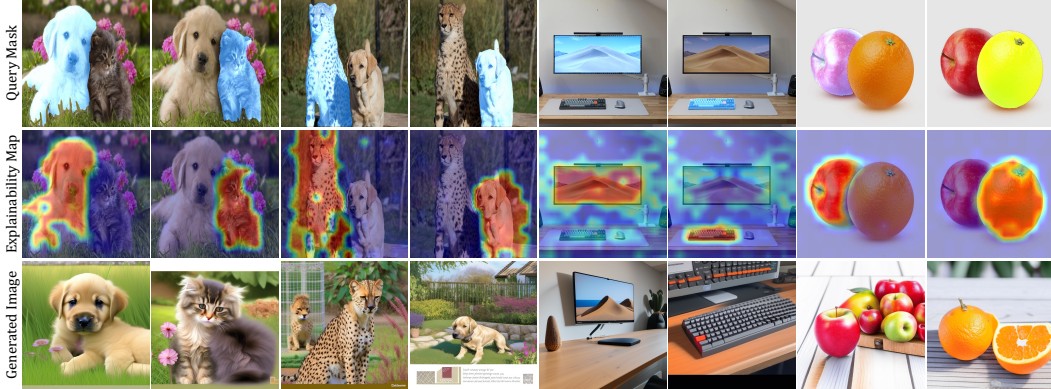

Figure 12: **Additional Localized Diffusion Examples.**

## M    ADDITIONAL LOCALIZED CAPTIONS

Figure showcases additional examples of localized captions for different masks as well as the final explainability map of the associated localized embedding. We observe that the generated caption essentially focuses on the area covered by the query mask, validating that the proposed MaskInversion is able to steer the visual focus toward the desired region.

## N    ADDITIONAL LOCALIZED DIFFUSION

Figure 12 provides additional visualization of the learned localized embedding for different mask queries. The visualization of the final explainability map is also provided. We observe that for each example the MaskInversion process is effectively able to steer the visual focus of the vision encoder toward the area of interest. Interestingly, when prompted with the mask of the monitor, the generated image contains a monitor with the same wallpaper scene, hence showcasing that the learned localized embedding learned a rich representation of the queried area.

