# OpenReview forum: "MaskInversion: Localized Embeddings via Optimization of Explainability Maps"
_ICLR.cc/2026/Conference — ICLR 2026 Poster_

### Official Review · Reviewer_dADG · 2025-10-26

**Soundness:** 3
**Presentation:** 4
**Contribution:** 3
**Rating:** 8
**Confidence:** 5

**Summary:**

This paper introduces MaskInversion, a novel and practical test-time optimization method for generating localized embeddings from pre-trained, frozen vision-language models like CLIP. The core problem it addresses is that models such as CLIP are trained for global image-text alignment (via the [CLS] token) and thus struggle with tasks requiring region-specific understanding.

MaskInversion works by initializing a new, learnable "Localized Embedding Token" (LETm) and iteratively optimizing it. The key insight is to use the model's own explainability map (e.g., LeGrad) as a guide. The optimization objective is to refine the LETm such that the explainability map it produces becomes spatially aligned with a given input query mask. This is achieved by minimizing a Dice loss between the explainability map and the mask. The resulting LETm can then be used as a "drop-in replacement" for the global [CLS] token in various downstream tasks, such as localized classification, referring expression retrieval, and even localized captioning and image diffusion.

**Strengths:**

* Novel and Elegant Method: The core idea of optimizing a new embedding by forcing its explainability map to match a target region is very clever. It's an elegant way to "invert" the model's attribution mechanism to achieve spatial control.
* Training-Free and Practical: The method works entirely at test time and keeps the powerful foundation model (like CLIP) completely frozen. This makes it extremely practical, versatile, and broadly applicable to any model that can produce a differentiable explainability map. It avoids the high cost and potential for catastrophic forgetting associated with fine-tuning.
* Strong and Comprehensive Evaluation: The paper demonstrates the effectiveness of MaskInversion across a wide array of tasks: referring expression retrieval (PhraseCut, RefCOCO), localized classification (PascalVOC, COCO, OpenImagesV7), localized captioning, and localized diffusion. The method consistently outperforms strong baselines, including those that modify the input (e.g., cropping, FGVP) and even those that require extensive fine-tuning (e.g., AlphaCLIP).

**Weaknesses:**

The original CLIP [CLS] token is powerful because it is aligned with text in a massive, open-vocabulary embedding space. It is not fully clear if the MaskInversion optimization, which pulls the token towards a spatial objective, fully preserves (or ideally, enhances) this fine-grained semantic alignment for general open-vocabulary classification. The regularization term ($L_{reg}$) helps, but this is a trade-off.

**Questions:**

Following on from the weakness, could the authors comment on the zero-shot generalization of the LETm? A valuable experiment, perhaps for the final version or future work, would be to use the ImageNet-S dataset (919 classes), which provides high-quality segmentation masks. How does the zero-shot classification accuracy of the MaskInversion LETm (using ground-truth masks) compare to the standard global [CLS] token on this large-scale benchmark? This would provide a definitive answer as to whether this localization technique consistently enhances or potentially trades off with raw zero-shot classification power.

**Details Of Ethics Concerns:**

no concerns

---

> ### Author Response · Authors · 2025-11-25
> **Reply to Reviewer dADG**
>
> # **Expected behavior on ImageNet-S**
>
> In standard zero-shot classification settings such as ImageNet-S, images typically contain a single dominant object that occupies most of it. In such cases, we anticipate that MaskInversion's localized embedding token [LET_m] would perform comparably to the global [CLS] token for two main reasons:
> 1. **Initialization and context:** The [LET_m] token is initialized from the [CLS] token, and when the mask covers most of the image (as is typical in ImageNet-S), our optimization objective (Eq. (4))  naturally aligns with the global image representation.
> 2. **Contextualized embeddings:** As illustrated in our localized diffusion and localized captioning experiments (**Section 4.5, 4.6, and Figure 3**), MaskInversion produces contextualized embeddings that retain background information relevant to the object. Therefore, the primary effect of MaskInversion in such scenarios would be to slightly reduce the influence of background, which should not degrade zero-shot classification accuracy.
>
> In terms of quantitative performance, we appreciate the reviewer's suggestion and we will include zero-shot classification experiments on ImageNet-S in the camera-ready version to provide definitive empirical evidence.

---

### Official Review · Reviewer_x9YP · 2025-10-28

**Soundness:** 3
**Presentation:** 3
**Contribution:** 3
**Rating:** 6
**Confidence:** 4

**Summary:**

The paper makes a valuable and well-executed contribution to the field of localized vision-language representation learning. It addresses a critical limitation of contrastive vision-language foundation models (e.g., CLIP) — their focus on global rather than regional alignment — with a method that balances effectiveness, efficiency, and flexibility. The work’s broad applicability across downstream tasks (referring expression retrieval, class retrieval, localized captioning, diffusion) further strengthens its impact.

**Strengths:**

1. Avoiding the "domain gap" of input-modification methods (e.g., RedCircle, Masked Crop) and the "data hunger" of fine-tuning methods (e.g., AlphaCLIP). By keeping the foundation model frozen and only optimizing an embedding token, MaskInversion retains the pretrained model’s knowledge while enabling task-agnostic localization.
The gradient decomposition strategy to reduce computational cost for multiple masks. This addresses a practical bottleneck of gradient-based explainability (expensive second-order derivatives) and makes the method scalable for real-world use cases.
3. The regularization loss (α) to balance local region details and global image context. This tunable tradeoff is absent in baselines and adds flexibility for tasks with varying context needs.

**Weaknesses:**

1. Upscaling increases the number of visual tokens (n), which affects the gradient decomposition’s efficiency (Equation 5 depends on n). The paper’s runtime ablation (Table 5) uses standard resolutions but not high-res inputs, leaving a gap in practicality for high-detail tasks (e.g., medical imaging).
2. For images with ≥5 objects, does MaskInversion’s performance degrade (e.g., due to cross-mask interference)? The current class retrieval and referring expression tasks focus on single masks, not overlapping or dense masks.

**Questions:**

1. In the gradient decomposition (Equation 5), you assume LETm(k)​ is independent of activations AL. Can you formally prove this independence, or provide empirical evidence that violating it (e.g., for highly complex masks) does not harm performance?
2. How does MaskInversion perform when masks overlap ? The Dice loss may struggle to distinguish overlapping regions, but this scenario is not tested.

---

> ### Author Response · Authors · 2025-11-25
> **Reply to Reviewer x9YP 1/2**
>
> # **W1 Runtime increases with the number of visual tokens**
>
> Theoretically, our gradient decomposition (**Eq. (5)**) has computational complexity linear in the number of visual tokens $n$ (due to the dot product operation). To empirically validate this, we conducted an additional experiment varying the input image resolution from 224 x 224 (196 tokens) up to 768 x 768 (2304 tokens). The results confirm that runtime indeed scales linearly with the number of visual tokens. We have added a figure with a discussion illustrating this linear relationship to the **Appendix Section E** of the revised paper (**marked in blue**).
>
> ---
> # **W2. Images with >=5 objects**
> MaskInversion optimizes each mask independently. For each query mask, we learn a separate localized embedding token [LET] through an independent optimization process. Consequently, the number of masks in an image does not affect the quality of individual embeddings—there is no cross-mask interference during optimization. The gradient decomposition strategy (**Section 3.2**) specifically exploits this independence to improve computational efficiency when processing multiple masks from the same image.
>
> # **Q2 + W2. How does MaskInversion perform when masks overlap?**
>
> Overlapping masks do not pose a problem for MaskInversion. Since each mask undergoes an independent optimization process, each overlapping mask is simply treated as another distinct query region with its own binary mask specification. The Dice loss is computed between the explainability map and the binary mask for that specific region, ignoring any other masks that might overlap with it. The optimization objective for each [LET] token remains well-defined and unaffected by the presence of other masks.
> Our experiments already validate the ability of MaskInversion to handle overlapping masks. For instance, the COCO dataset has ~8 masks per image on average, some of which overlap or are in close proximity. Despite this, MaskInversion achieves better performance than other methods on various tasks involving COCO (Tables 2 and 3). Similarly, in our multi-object analysis (**Appendix Section G, and Fig. 6**), we demonstrate qualitatively that MaskInversion successfully handles masks covering multiple objects or overlapping regions.
>
> We have updated **Section 3.2** in the paper (**marked in blue**) with details about this matter to avoid potential confusion.

---

> > ### Author Response · Authors · 2025-11-25
> > **Reply to Reviewer x9YP 2/2**
> >
> > # **Q1 Can you prove that LETm(k) is independent of activations?**
> >
> > We appreciate the reviewer's scrutiny regarding the gradient decomposition in Equation (5). We clarify here that **$LET_\mathbf{m}^{(k)}$ is indeed independent of the activations $\mathbf{A}$ during the computation of the explainability map gradient.**
> > Conceptually, while $LET_\mathbf{m}^{(0)}$ may be initialized using values from the model (the [CLS] token), in our implementation this token is **detached** from the computational graph immediately upon initialization. It is treated as an external, standalone optimization variable (similar to a learnable query vector or a weight parameter in a linear layer) rather than a dependent function of the image input.
> > Below, we provide the formal proof demonstrating why this independence holds and why the decomposition is exact.
> >
> > ### Formal Derivation
> >
> > Let the scalar score function be defined as the dot product between the global image representation $\bar{\mathbf{z}}$ and the localized embedding token $LET_\mathbf{m}^{(k)}$ (omitting normalization for clarity, as it does not change the dependency logic):
> > $$
> > s = \bar{\mathbf{z}} \cdot (LET_\mathbf{m}^{(k)})^T
> > $$
> > where:
> > 1.  $\bar{\mathbf{z}} = \frac{1}{n} \sum_p z_p := f(\mathbf{A})$: represents the combined patch and [CLS] token representation averaged across the spatial dimensions which is a function of the activations $\mathbf{A}$.
> > 2.  $LET_\mathbf{m}^{(k)}$: The learnable vector at iteration $k$.
> >
> > We want to derive the gradient of $s$ with respect to the activations $\mathbf{A}$, denoted as $\nabla \mathbf{A}$. Applying the product rule, we obtain:
> >
> > $$
> > \nabla \mathbf{A} = \frac{\partial s}{\partial \mathbf{A}} = \frac{\partial (\bar{\mathbf{z}} \cdot (LET_\mathbf{m}^{(k)})^T)}{\partial \mathbf{A}}
> > $$
> >
> > $$
> > \frac{\partial s}{\partial \mathbf{A}} = \left( \frac{\partial \bar{\mathbf{z}}}{\partial \mathbf{A}} \cdot (LET_\mathbf{m}^{(k)})^T \right) + \left( \bar{\mathbf{z}} \cdot \frac{\partial (LET_\mathbf{m}^{(k)})^T}{\partial \mathbf{A}} \right)
> > $$
> >
> > To evaluate the second term, we examine the definition of $LET_\mathbf{m}^{(k)}$. This token is an optimization variable updated via an external loop. Even in the case where we initialize $LET_\mathbf{m}^{(0)}$ using the [CLS] token derived from $\mathbf{A}$, we perform a "stop-gradient" operation (detach) at initialization:
> > $$
> > LET_\textbf{m}^{(0)} := \text{stop-gradient}(\text{CLS-Token})
> > $$
> > Because $LET_\mathbf{m}^{(k)}$ is a detached set of parameters and not a continuous function of the current forward pass activations $\mathbf{A}$, its partial derivative with respect to $\mathbf{A}$ is zero:
> > $$
> > \frac{\partial LET_\mathbf{m}^{(k)}}{\partial \mathbf{A}} = \mathbf{0}
> > $$
> > Consequently, the second term of the expansion vanishes:
> > $$
> > \nabla \mathbf{A} = \left( \frac{\partial \bar{\mathbf{z}}}{\partial \mathbf{A}} \cdot (LET_\mathbf{m}^{(k)})^T \right) + \left( \bar{\mathbf{z}} \cdot \mathbf{0} \right)
> > $$
> > $$
> > \nabla \mathbf{A} = \frac{\partial \bar{\mathbf{z}}}{\partial \mathbf{A}} \cdot (LET_\mathbf{m}^{(k)})^T
> > $$
> >
> > This confirms that Equation 5 in the paper is mathematically exact under the definition that $LET_\mathbf{m}^{(k)}$ is a learnable parameter independent of the current forward pass, regardless of whether it was initialized randomly or via a detached copy of the [CLS] token.
> >
> > We have added this derivation to **Section I of the Appendix** of the updated version of the paper.

---

> > > ### Comment · Reviewer_x9YP · 2025-11-26
> > >
> > > Thank you for the detailed response and thorough explanations. The authors have addressed most of my concerns, and I have no further comments. Based on the feedback, I decided to maintain my rating.

---

### Official Review · Reviewer_dTuU · 2025-10-31

**Soundness:** 3
**Presentation:** 2
**Contribution:** 2
**Rating:** 4
**Confidence:** 3

**Summary:**

This paper proposes a method called MaskInversion to obtain local representations for masked query objects during test time by optimizing the objects' heatmaps and the provided binary masks. MaskInversion can be used for obtaining such local object embeddings generally from different pre-trained foundation models and the learned local embedding can be used as a drop-in replacement for improving downstream region-based tasks. The authors conduct experiments on downstream tasks including referring expression retrieval, class retrieval for segmentation. They compare with other training-free methods and show consistent improvements.

**Strengths:**

(1) MaskInversion is training-free can be used generally for region-based retrieval tasks.

(2) The authors provide many ablation studies to validate the proposed method.

**Weaknesses:**

(1) In the paper, the authors claim MaskInversion can be used for any pre-trained model. Have the authors conduct experiments on any vision-language models such as llava?

(2) In the section 4.2, the authors mention that for optimization of MaskInversion, they set "10 gradient descent iterations". I'm curious how this number of iterations is decided?

(3) Have the authors try other loss functions such as simple MSE loss?

**Questions:**

See weakness

---

> ### Author Response · Authors · 2025-11-25
> **Reply to Reviewer dTuU**
>
> # **W1 Experiments on models such as LLaVA.**
> We thank the reviewer for this important clarification question. MaskInversion is designed to generate localized embeddings and therefore requires the model to natively produce embedding representations, such as contrastive models like CLIP. Models such as LLaVA are generative vision-language models that are trained for text generation rather than embedding extraction, and thus do not directly output embeddings.
>
> However, MaskInversion can be applied to the vision encoder component of such generative models. We demonstrate this in our localized captioning experiments (*Section 4.5, Table 6*), where we apply MaskInversion to CLIPCap—a captioning model that uses CLIP as its vision encoder. By replacing the global [CLS] token with our localized embedding tokens [LET], we successfully generate captions focused on specific masked regions without any fine-tuning of the model.
>
>
> ---
> # **W2 Number of iterations K**.
> The value of $K=10$ gradient descent iterations was determined through empirical analysis balancing convergence quality and computational efficiency.
> We have updated the paper to include a convergence analysis figure, in the **Appendix Section D**, that illustrates this choice. The figure shows both the loss (left axis) and accuracy on PascalVOC (right axis), demonstrating rapid convergence within the first 10 iterations. Specifically, the accuracy reaches approximately 85% by iteration 10 and then plateaus. This behavior reflects that the localized embedding token [LET] converges quickly, with the explainability map becoming sufficiently aligned with the query mask within 10 iterations. While the loss continues to decrease marginally beyond 10 iterations, these residual improvements primarily reflect boundary refinement between the explainability map and mask. Our ablation experiments on mask quality (**Table 4**) demonstrate that this has minimal impact on the semantic quality of the learned embedding. Therefore, $K=10$ iterations provides a good balance between achieving a well-localized embedding and maintaining computational efficiency.
>
> We are happy to move the figure and the discussion from the **Appendix Section D** to the main paper upon the reviewer’s request.
>
> ---
> # **W3 Other loss functions.**
> We conducted early experiments comparing the Dice loss with alternatives, including the MSE (L2) loss and the Binary Cross-Entropy loss. We chose the Dice loss because it is a standard metric for measuring region similarity in segmentation tasks and has well-established properties for optimizing spatial alignment [1]. In our early experiments, we found the Dice loss to outperform both MSE and Binary Cross-Entropy, converging significantly faster while achieving higher accuracy on downstream tasks such as class retrieval on PascalVOC. The superior performance of the Dice loss can be attributed to its design for directly measuring overlap between two regions (the explainability map and the query mask), making it particularly well-suited for our mask alignment objective. In contrast, MSE penalizes spatial misalignment and intensity differences equally, which is less suitable for guiding the embedding token toward the correct image region.
> Please indicate if we should rerun experiments with the alternative losses before the end of the discussion
>
>
> [1] MILLETARI, Fausto, NAVAB, Nassir, et AHMADI, Seyed-Ahmad. V-net: Fully convolutional neural networks for volumetric medical image segmentation. In : 2016 fourth international conference on 3D vision (3DV). Ieee, 2016

---

### Author Response · Authors · 2025-11-25
**General Response**

We thank the reviewers for their constructive comments and positive assessment of our work.
We are encouraged that **Reviewer dADG** found our approach to be a _"novel and elegant"_ solution with a _"strong and comprehensive evaluation"_ across various tasks. We also appreciate **Reviewer x9YP**'s recognition of the method's computational efficiency via gradient decomposition and its ability to retain the foundation model's knowledge without the domain gap introduced by input modifications. Finally, we thank **Reviewer dTuU** for valuing our extensive ablation studies and the general applicability of our training-free framework for region-based retrieval. We have carefully addressed all questions and updated the manuscript accordingly. Below, we provide detailed responses to specific points.

---

### Meta-Review · Area_Chair_FisF · 2026-01-05

**Summary:**

This paper proposes a training-free method for extracting localized information from global image encoders such as CLIP from specified query masked regions. Without involving any training of the original vision foundation model, the proposed approach uses explainability methods to constrain the explained region to the overlap with the masked query region and optimize a derived latent code from the image to to do. They also propose optimizations of the gradient-based optimization technique employed to align the vision token embedding to the explained query masked region. The authors show improved accuracy on many downstream localized inference tasks.

Three reviewers provided ratings of 4, 6, and 8, with two reviewers leaning positive and one championing the work. Most the of the reviewers' concerns were addressed in the rebuttal phase, including all of the concerns of the one reviewer that leaned towards rejection.

All things considered the AC feels that the paper makes a valuable and insightful contribution to the research community and hence recommends acceptance. The authors should include in the final version of the paper the changes that they have promised in the rebuttal.

Congratulations!

**Reviewer Concerns:**

Concerns addressed:
* Experiments on any vision-language models
* How the number of training iterations (10) are decided
* Scaling with higher resolution images
* Explaining the interactions of overlapping masks
* Proving that LETm(k) is independent of activations

Concerns not addressed:
* Experimental results on ImageNet-S dataset (919 classes)

**Reviewer Scores:**

1. Reviewer dTuU (Rating: 4: marginally below the acceptance threshold. But would not mind if paper is accepted)

The reviewer's primary concerns were (a) experiments on any vision-language models such as llava and (b) how the number of iterations for gradients is decided and (c) use of MSE loss versus the DICE loss. The authors have provided additional explanations and experimental results to clearly address all the reviewer's concerns. Hence, the reviewer is likely to have increased their original score.

2. Reviewer x9YP (Rating: 6: marginally above the acceptance threshold. But would not mind if paper is rejected)

This reviewer's concerns were around (a) applicability to high-resolution tasks where the number of token $n$ is larger and (b) how the method works when masks overlap. The authors' rebuttal address their concerns and they maintained their original positive rating.

3. Reviewer dADG (Rating: 8: accept, good paper (poster)

This reviewer's primary concern was if the method of fine-grained semantic alignment works also for general open-vocabulary classification or hampers it. While the authors provided speculative discussions of how this might play out, they did not provide experimental results on the ImageNet-S dataset (919 classes) as the reviewer had requested. They have promised to include this in the final paper, if it is accepted.

---

### Decision · Program_Chairs · 2026-01-26

Accept (Poster)